# Equine Hepacivirus: A Systematic Review and a Meta-Analysis of Serological and Biomolecular Prevalence and a Phylogenetic Update

**DOI:** 10.3390/ani12192486

**Published:** 2022-09-20

**Authors:** Giulia Pacchiarotti, Roberto Nardini, Maria Teresa Scicluna

**Affiliations:** National Reference Center for Equine Diseases, Istituto Zooprofilattico Sperimentale del Lazio e della Toscana “M. Aleandri”, via Appia Nuova 1411, 00178 Rome, Italy

**Keywords:** Hepacivirus, equids, systematic review, risk factors, meta-analysis, phylogenesis

## Abstract

**Simple Summary:**

This is a comprehensive review containing the most up-to-date information on Equine Hepacivirus, one of the recently discovered hepatic equine viruses, together with an analysis of serological and biomolecular presence presented in apreviously published papers, and an update on its genetic relationship within the species and with similar species. Extensive description of the EqHV features is included, and results are presented with several tables and figures, providing a valuable reference guide for further studies.

**Abstract:**

Viral hepatitis has recently assumed relevance for equine veterinary medicine since a variety of new viruses have been discovered. Equine Hepacivirus (EqHV) is an RNA virus belonging to the *Flaviviridae* family that can cause subclinical hepatitis in horses, occasionally evolving into a chronic disease. EqHV, to date, is considered the closest known relative of human HCV. EqHV has been reported worldwide therefore assessing its features is relevant, considering both the wide use of blood products and transfusions in veterinary therapies and its similitude to HCV. The present review resumes the actual knowledge on EqHV epidemiology, risk factors and immunology, together with potential diagnostics and good practices for prevention. Moreover, adhering to PRISMA guidelines for systematic reviews a meta-analysis of serological and biomolecular prevalence and an updated phylogenetic description is presented as a benchmark for further studies.

## 1. Equine Viral Hepatitis: Overview

Hepatitis is an inflammation of the liver which can occur by toxic, autoimmune, or infective causes. Among the latter, viral hepatitis has recently assumed greater relevance for equine clinical practice as a variety of new viruses have been discovered, such as the Non-primate Hepacivirus (NPHV, now classified as EqHV) [1], the equine pegivirus (EPgV) [2], the Theiler disease-associated virus (TDAV) [3], the equine parvovirus-hepatitis (EqPV-H) [4], and the Equine hepatitis B virus (EqHBV) [5]. This review presents current knowledge concerning EqHV infection in horses, including its epidemiology, evolutionary characteristics, diagnostics, and prevention practices.

## 2. Equine Hepacivirus (EqHV)

### 2.1. Classification

EqHV belongs to the *Flaviviridae* family, which includes more than 60 species [6], and consists of four genera: *Flavivirus*, *Pestivirus*, *Pegivirus*, and *Hepacivirus*.

Flavivirus is the largest genus, including arthropod-borne viruses infecting mosquitoes or ticks (West Nile virus, Usutu virus, yellow fever virus); pestiviruses infect mainly pigs and ruminants (classical swine fever virus, border disease virus, bovine viral diarrhea viruses 1 and 2); pegiviruses cause persistent infections in a wide range of mammalian species; and finally, hepaciviruses are known to cause chronic liver disease in humans (Hepatitis C virus) [7].

EqHV, a member of the latter genus, was discovered by [8] who, while researching viral flora in domestic animals, identified several flavivirus sequences in respiratory samples of dogs that had been implicated in respiratory illness outbreaks in the USA. Phylogenetic analysis on these samples revealed that the sequences belonged to a unique virus, strongly related to HCV, which was initially named Canine Hepacivirus (CHV). Since 2011, several other animal species have been tested, leading to the discovery of a whole new group of hepaciviruses in a number of species (Table 1). These viruses display different degrees of homology with HCV, with EqHV revealed to be the closest known relative of human HCV to date, having horses as its natural host [1].

### 2.2. Viral Structure

EqHV is a virus with spherical virions measuring 40–60 nm (Figure 1) and a lipid envelope characterized by a single, small capsid protein and two different envelope glycoproteins (E1, E2), organized in icosahedral units [6,7,8,9]. Whether the viral protein composition of the virions of most *Flavivirus* is known (the general structure is an icosahedral array of 90 envelope glycoprotein heterodimers forming a smooth surface), the same knowledge is not available for other members of the *Flaviviridae* family, which shows a lack of proteomic analysis data. However, recent studies reported that host and tissue cells influence the final composition of the virion envelope: the actual functions of these accessory proteins included during the budding and released from the host cell, if present, needs further study [10].

### 2.3. Viral Genome

Similar to other hepaciviruses, EqHV is a single-stranded positive-sense RNA virus, with an RNA of approximately 9500 bp [11] of which 9200 bp belong to the polyprotein gene [1,8]. In Figure 2, a graphical description of the genome is presented.

The EqHV genome presents a large open reading frame (ORF) encoding a multifunctional polyprotein, which is processed by host- and viral-encoded proteases into 10 distinct proteins: 3 structural (core, E1, and E2) and 6 non-structural (NS2, NS3, NS4a, NS4b, NS5a, and NS5b) [13]. The internal ribosome entry site (IRES) is similar in sequence and predicted structure to that of HCV and is capable of driving translation of the downstream ORF [11]. Glycosylation sites on the encoded envelope proteins (E1/E2) help to mask virions from the potential binding of host neutralizing antibodies: EqHV has the second-highest number of predicted glycosylation sites within the hepaciviruses, after HCV. Therefore, the potential correlation existing between the number of predicted glycosylation sites within hepaciviral genomes and rates of establishing persistent infection in vivo is object of study [13]. Another significant aspect of the genetic structure of EqHV genome is that, similar to other hepaciviruses which have been fully characterized, it also possesses putative seeding sites in the 5′ untranslated region (UTR) for the binding of host-expressed microRNA-122 [13,14]. MicroRNAs (miR) are small noncoding RNAs that have emerged as major regulators of gene expression and typically bind to the 3′ UTR of mRNAs through complementary base-pairing interactions [15]. The presence of these sites is a highly relevant characteristic when studying tropism and the virus life cycle; in fact, it has been recently reported that EqHV could also replicate in tissues different from the liver, but the presence of the binding miR-122 site on the 5′ UTR highlights the possibility of a selective force on hepatotropism to exploit the tolerant liver environment and to orchestrate chronicity [16]. Speculation can be made that because of its role in hepaciviruses infections, miR-122 could be used as a marker of acute hepacivirus-induced liver damage in horses since it appears more sensitive compared to other liver markers [17].

Overall, then, the presence of miR-122 would make EqHV completely hepatotropic, on an evolutionary preference pattern, as much as its human homologue, and it would be in agreement with several reports on horses since 2011 that highlighted the emergence of a disease associated with variation in liver enzyme values [1,8,11,17,18,19,20,21,22,23] and the ability to establish chronicity in horses, similar to what happens for HCV in humans [11,18,19,20,23,24,25,26].

### 2.4. Viral Replication

The general life cycle and protein interactions of hepaciviruses have been thoroughly investigated; in this chapter, only a general summary will be reported (for a more in-depth focus, refer to [10]).

EqHV penetrates the host cell by receptor-mediated endocytosis, triggered by the attachment of its glycoproteins to the host cell surface. The main adhesion factors are heparin sulfates, dendritic cell-specific intracellular adhesion molecule (ICAM), and C-type lectin receptors (e.g., DC-SIGNR, expressed in endothelial cells). Clathrin-dependent uptake has been described as the major endocytosis mechanism, not excluding the possible presence of alternative strain-specific entry routes.

Fusion within endosomal compartments is the second step of the entry process: usually it requires a low pH trigger to allow the viral surface glycoproteins to undergo a conformational change, but the whole mechanism is driven by unknown factors in hepaciviruses. The last entry step is the uncoating of the viral capsid, which eventually delivers the RNA to the cytoplasm: this, as well, is one of the least studied processes of the virus life cycle. What is known, however, is that protein interactions of the virus with the host cell may hamper early antiviral signaling and interferon production, therefore providing hints of viruses strategies to impede innate immune recognition. Overall, though, these primary steps are still almost completely unclear.

The processes of translation, genome replication, and particle assembly take place in the endoplasmic reticulum (ER) which is remodeled during infection. Translation happens in host ribosomes, and replication requires viral non-structural proteins (NS) and several host proteins. The viral genome is translated in a single polyprotein that is cleaved by proteases and separated in functional subunits. NS3, NS4, and NS5 rearrange the membranes of the ER in order to create compartments in which the virus can replicate itself and subsequently assemble new virions. Protein interactions, in these steps, can induce changes even to the mitochondria architecture, necessary to prevent viral RNA recognition by the innate immune system. The last step is the assembly of new viruses, the budding, and their release outside the host cell. This step can involve structural and translated proteins that do not necessarily originate from the viral RNA and that are included into a new virion. NS proteins play a role in this process, probably helping in delivering newly replicated viral RNA into new particles and in recognizing early RNA compared to later stages of replicated RNA. The whole particle is then assembled and buds into the ER, acquiring a lipid envelope and the structural characteristic glycoproteins. Virions are finally released through the typical secretory pathway. The specific composition of each virion may highly differ depending on the host and tissue cell in which the assembly has taken place. To date, in fact, there is evidence that similarly to how each cell type expresses its unique proteome, virions’ surface composition can be affected by the specific composition of the subcellular site of virion release.

## 3. Epidemiology

### 3.1. Routes of Transmission

#### 3.1.1. Vertical Transmission

To date, vertical transmission has been recorded in two single studies [27,28]. In the first study, 21 Thoroughbred mares were monitored before parturition and for 6 months post-partum. Among them, four mares carried EqHV RNA at parturition and one of the foals was also EqHV RNA positive after birth [27]. In the other study, out of 394 dead foals or fetuses, 3 (0.76%) showed the presence of the EqHV genome. EqHV was also detected, for the first time, in two allantochorions: one of them had a 100% sequence homology with that identified in the mare’s serum [28]. This evidence suggests a clear vertical transmission route, even though the authors also hint that probably only variants of the virus could promote EqHV in utero transmission since, otherwise, the numbers of cases worldwide should be much higher than observed. Additionally, they suggested that the allantochorion tissue itself could have been a site of replication of the virus and the source of transmission from the mare to the fetus (since the equine epitheliochorial placenta has six layers of maternal and fetal tissues between the two blood circulations, making it difficult for viruses to cross that barrier) [28].

#### 3.1.2. Horizontal Transmission

Studies on horizontal transmission have been recently conducted, showing however a lack of strong consistence. Some evidence of the EqHV genome in nasopharyngeal swabs were, in fact, detected [29,30,31]. In addition, in Gather et al. (2016) [27] some foals developed EqHV infection during the 6 months of monitoring that followed parturition, along with the other two mares: a part from one foal which received plasma transfusion 1 day post-partum with plasma that tested EqHV RNA positive (making it the likely route of infection), the infection of the other foals and mares suggests the presence of an alternative and probable horizontal route of transmission. Therefore, more studies are needed to elucidate if transmission involving the oropharyngeal sphere is possible.

For what concerns fecal–oral transmission of EqHV, instead, a recent study suggests that there is no evidence that it could be a source of transmission [32].

#### 3.1.3. Parenteral Transmission

Parenteral transmission of EqHV has been demonstrated by [11,17,24]: in their studies, infections were performed in order to study EqHV features and all experimental infections were associated with acute or chronic liver pathology, accompanied with an increase in viral genome copies during the first week post infection until the peak of infection [11,17,21,24,25,33]. Iatrogenic transmission through inoculation of biological products, such as vaccines and plasma, or through the reuse of needles due to inattentive veterinary practices could have then been a major source of the spread before 2012, when EqHV had not yet been discovered and effectively characterized [1]. Similar experiments were also undertaken later and demonstrated to associate injection of RNA positive sera with the effective development of subclinical hepatitis also in individuals who had already been infected with the same or a different hepacivirus [17,21].

##### The Relevance of Parenteral Transmission, Horse Serum, and Plasma

Blood derivates are widely employed both for research purposes and in clinical practice. In research, many cell culture systems, vaccines, and the production of anti-sera depend on serum: the main product used in biotechnology is usually fetal bovine serum (FBS), but its frequent contamination with ruminant pestiviruses (family: *Flaviviridae*) highlighted several issues that, to be overcome, resulted in attempts to adapt cell lines to equine sera [34,35]. Since 2012, many authors have searched for the presence of different viruses and *Flaviviridae* specimens in particular also in commercial equine serum products commonly used for the production of anti-sera, antitoxins, cell culture propagation, vaccines, and pregnant mare serum gonadotropin (PMSG) [1,11,34,35,36,37].

When Burbelo et al. in 2012 [1] first described EqHV in horses, among their serum samples there were also commercial horse serum pools and one of them, from New Zealand, had a positive result for EqHV: its sequence showed identity with the previously discovered CHV [8], suggesting the possibility of EqHV of a spillover from horses to dogs, posing a high threat for veterinary horse product. The hypothesis of a cross species transmission is supported by the fact that other studies [38,39] report dogs that had come into regular contact with EqHV positive horses and tested positive themselves, or seropositive. Speculations can be made that either vaccine products contaminated with EqHV positive horse sera, food (offals given as food), or horizontal transmission caused by sharing the same environment could be the sources of infection in dogs [11,17,19,21,27,29,30,31,39,40,41].

A wider analysis specifically on commercial horse serum products was provided in 2015, when among 15 sera from different companies, 14 tested positive for EqHV RNA [11]. After the demonstration that transmission of EqHV by plasma transfusion is possible and that hepatitis can be induced in horses [24], further interest grew around the presence of viral genomes in equine blood products used in veterinary practice. Similar results regarding the high presence of EqHV in commercial horse sera were obtained in the following years [34,35,36]; however, to date, no viral genome has been detected in PMSG products [36].

Considering the biological relevance of equine blood products both for research and for therapeutic purposes, biosecurity and innocuity of these products should be of primary relevance to avoid the spreading of the virus through veterinary practice and interference with research studies that employ equine-derived supplement [11,34,35,36,37].

#### 3.1.4. Other Routes of Transmission (Insect-Mediated and Sexual)

Even though the majority of members in the genus *Flavivirus* are arthropod-borne [6], there is a general lack of information about EqHV transmission by insects or arthropods. In the very first study of surveillance of EqHV in Austria [24], a sample of 5338 individuals from 10 species of mosquitos was tested and no viral RNA was detected. This model should be applied to all geographical areas and within all species of possible vectors, to certainly assess the possibility of a vectorial route of transmission of the virus. Moreover, it remains to be determined whether EqHV can be mechanically transmitted by other hematophagous insects or arthropods [40].

Transmission through sperm, instead, has not been reported: more studies are needed to possibly support the feasibility of this transmission route [28], especially because of the high relevance of certain genetic traits conferred to purebreds that travel expressly in order to breed. If demonstrated, it could change the approach on selective breeding and it would highlight the necessity for more attentive management practice at least for high value individuals.

## 4. Geographical Distribution, Viral Prevalence, and Seroprevalence

EqHV infections in horses have been recorded worldwide, from Europe [18,19,29,33,34,38,42,43,44,45,46,47,48] to Africa [39,49], from America [1,22,50,51,52] to Asia [26,32,53,54,55,56,57,58,59,60] and even Oceania [61]. Details on the features of each study are presented in Table 2.

This chapter will focus on the results of a meta-analysis on EqHV prevalence (virological and serological) worldwide, updated to June 2022.

### 4.1. Meta-Analysis

#### 4.1.1. Search Strategy and Study Selection

A systematic literature review and a meta-analysis approach were performed to collect studies related to the serological and biomolecular prevalence of EqHV worldwide, following the Preferred Reporting Items for Systematic Reviews and Meta-Analyses (PRISMA) guidelines [62]. A systematic search in the databases PubMed, Scopus, and Science Direct was performed to scan for all studies within the scope of our subject. Since EqHV has been named differently from it first detection throughout time (CHV, Non Primate Hepacivirus, Equine Hepacivirus), in order to carry out a comprehensive research, all known names of the virus were included in the keywords used: the key terms were searched in the title, in the abstract, and/or keywords of studies. The use of ‘equine’ and ‘horse’ in the key terms made it possible to extract studies conducted specifically on horses and quotation marks (‘’) were used to include studies using the entire relevant expression. The following key terms were used to select studies on EqHV: (Equine Hepacivirus) or (Non Primate Hepacivirus) AND (Prevalence); (Horse) AND (Hepacivirus); (Canine Hepacivirus).

EqHV being a relatively new virus, journal articles and short communications were selected with no restriction on the publication date. Book chapters were excluded.

The citations of the identified studies containing the title and abstract were screened in the Mendeley bibliographic manager, duplicated works were excluded and the titles and abstracts were read. After a first round of selection based on titles and abstracts, a second selection was carried out based on a detailed review of the full text of the studies. Those involving countries from all around the world and reporting serological and/or biomolecular prevalence of EqHV were kept for this review. To be considered effective and to be included in the review, only studies on the prevalence that counted a minimum of 100 individuals and focused not on symptomatic horses were selected.

In detail, [19,22,53,58] were excluded for the number of samples <100; refs. [33,48] were excluded because the sample employed seemed to be the same as [38,42], respectively. Because the studies present data on infection trials and not seroprevalence studies, refs. [11,17,21,24] were excluded. In Figure 3, the PRISMA flowchart is presented.

#### 4.1.2. Data Extraction

From the eligible studies, data extraction included (i) the location (region and country), (ii) the species involved, (iii) the total number of equids included, (iv) the total number of positive, and (v) the diagnostic technique used/PCR target.

#### 4.1.3. Meta-Analysis

The papers selected by the systematic literature review were then used to perform a meta-analysis on biomolecular and serological prevalence in horses to quantify and compare the level of circulation of EqHV in equids and to estimate a proxy of prevalence and seroprevalence in the world. For each study, prevalence was calculated with 95% confidence intervals (CIs).

Then, meta-analysis on both prevalence and seroprevalence in equids was performed using a random model effect (REML method), using the DerSimonian–Laird method. The publication bias was verified by the funnel plot, which is a scatter plot of the effect (prevalence) on the *x*-axis and a measure of the study’s precision (inverse of standard error) on the *y*-axis, and the asymmetry of this graph may suggest publication bias. The Rank correlation test and the Egger statistical test were used to confirm or not the asymmetry of the funnel plot. Moreover, according to [63] selection model, PET PEESE analysis, and Robust Bayesian Meta-Analysis (RoBMA) were performed to further confirm publication bias. All statistical analyses were performed using JASP software version JASP (version 0.16.3, University of Amsterdam, Amsterdam, The Netherlands).

In Figure 4 and Figure 5, the results of meta-analysis on seropositivity and biomolecular positivity are respectively reported.

## 5. Individual Risk Factors

Even though definitive data are not available, it has been reported that certain breeds, individuals, and production areas are particularly susceptible when studying EqHV prevalence: for example, young horses [25,39,46,49,52,54,56], Thoroughbreds [19,34,45,46,49,54,56], competition horses [50,54,55,58], and females [52,59]. In this chapter the main risk factors: age, breed, sex, and production category will be examined.

### 5.1. Age

A general trend of higher infection rates in juvenile horses (younger than 8 years) have been reported [25,39,46,49,52,54,56]. Younger horses seem to be more susceptible, probably due to their immune system still developing as it takes up to several months to guarantee a certain protection. When foals with severe combined immunodeficiency (SCID) were experimentally inoculated as part of a study to investigate if EqHV and horses could be an in vivo model for HCV, they presented minimal to absent disease, compared to adults [24]. Since clinical disease in immunocompetent foals was also mild or absent, it can be speculated that adaptive immunity contributes in generating EqHV-associated liver damage, since SCID foals lack functional B and T lymphocytes, which have been indicated as being potentially co-responsible for the emergence of the disease and liver damage in horses older than 6 months [17,21]. Of note, both immunocompetent foals and SCID foals were not able to clear EqHV infection, whereas young adult horses did clear the virus, implying other still unknown immune pathways are involved in conveying immunity. However, since neither the potential infectious routes, nor the development of the immune system of young horses have been fully characterized yet, there is still a great uncertainty about the role age plays in EqHV infections: this is also due to other reports that seem to suggest the opposite, making older horses (older than 10 years) more susceptible [32,59].

### 5.2. Breed

In several studies, Thoroughbreds showed higher viremic prevalences worldwide, compared to other tested breeds [19,34,45,46,49,54,56]. Thoroughbred horses are known for their use in horse racing and equestrian sports since their main skills are considered to be athleticism, agility, and speed; when retired, they usually become family horses, but they maintain a high economical and personal value. Thoroughbred horses are often expensive and hardly replaceable, hence great attention is focused on their health, pathologies, and well-being. The higher prevalence encountered in this breed, then, could be either related to genetic factors or ascribable to the variety of preferential treatments they are subjected to, especially international transportation (for both races and breeding) and vaccine therapies, making them naturally more exposed to potential sources of infection [19].

### 5.3. Sex

Females seem to be more viremic compared to males when both sexes are screened, but also this trend is debatable. Few reports [39,52,59] underline how females show higher viremic prevalences compared to males, and male prevalences are higher in other studies [51]. The reason behind the higher prevalence in one of the two sexes could be due to transportations and encounters with other horses in order to breed [32].

### 5.4. Production Category

Competition and reproduction horses share similar risk factors: they are both considered of economic interest and are compelled to travel and come in direct contact with other animals from different stables and, often, countries or even continents. As a matter of fact, a few studies report that compared to other production categories, competition horses seem to have a higher viremic prevalence [50,54,55,58], hinting that probably stress induced by transportation, frequent contact with other horses, and promiscuity, or lack of hygienic hosting structures might be the origin of the prevalence values encountered in these categories.

## 6. Herd Management Risk Factors

Poor performance can be greatly counter-productive, especially when referring to horses with a high economical value, such as racehorses; hence, good management practices are fundamental to guarantee health and performance.

It is well known that horses are sensitive to environmental changes, social perturbations, transportation, and stable density: all of the above can result in an increase in stress indicators, such as increased heartbeats, cortisol levels, dehydration, hemoconcentration, and reinforcing the display of stress behaviors in conjunction with stressful conditions [46,64,65,66,67,68].

Immunosuppression induced by stress is a renowned fact in horses [46,65,68,69] and therefore these factors become relevant and potential concurrent causes when studying horses diseases. A weakened immune status could be implied in a more intense clinical manifestation of diseases in certain animals: veterinary treatments, change in herd managements, international transportation, direct contact during races and events, and stable changes in order to allow controlled breeding could expose vulnerable individuals to stressing situations, and this could increase the susceptibility of certain horses (also considering the importance of personality traits each animal shows and individual vulnerability to stress-induced responses).

These factors indeed deserve further in-depth analysis in order to identify whether they are more or less significant when studying pathologies connected to the diagnosis of EqHV in horses and to, eventually, lead to meliorative management practices. These aspects, a part from a higher attention on the health of the animals (vaccine, therapies, and routine screenings), should be one of the main focuses of future management protocols in order to make events and competitions safer for the animals and to defend owners’ interests.

Of note, there is currently no evidence that EqHV could pose a threat to human health, even when referring to handlers in prolonged contact with potentially infected animals [26,70].

## 7. EqHV Clinical Presentation

### 7.1. Hepaciviruses Infection: A Brief Summary

EqHV is not only similar to HCV from a genetic point of view but also in the symptoms it causes in horses: in fact, it is responsible for subclinical hepatitis and it can sometimes degenerate into a chronic disease, similar to what happens in humans [11,17,18,20,23,24,26,27,70,71].

In 2015, Ramsay et al. [24] provided, for the first time, complete data on the replication kinetics of a EqHV infection in vivo, along with proof of associated liver disease in horses after experimental transmission with plasma containing the EqHV genome. Similarly, in 2015, in 2017 and 2021 other authors proposed comparable studies [17,19,21] that highlighted the main trends around which a EqHV infection is oriented: first, is the evolution of the viremic load, second the increase in NS3 antibodies (Ab) at seroconversion, and third the variation in concentration of specific liver enzymes as the gamma-glutamyl transferase (GGT) and the sorbitol dehydrogenase (SDH), which are apparently the most significant in terms of variation compared to the reference intervals, when analyzing EqHV infections (Table 3). The infection, then, diversifies depending on acute or chronic progression and eventually is cleared.

Overall, the standard gait of EqHV infection usually begins soon after the infection, even within 1 week post infection (WPI) with relatively high levels of detectable RNA copies/mL (10^5^ to 10^8^ viral RNA copies/mL at the peak) and the viremic status lasts several weeks (15–30), with a slight decrease in viremia after approximately an average of 15 weeks. Soon after the viremia peak, higher levels of SDH and GGT can be observed as well, 1.5× to 5.0× times above the upper limit of normal reference ranges: they remain high until approximately 25 WPI and tend to decrease 2 weeks after reaching the peak values.

These fluctuations also apparently have repercussions on liver tissue since mild hepatocellular damage could be observed at the same time through biopsy: hepatic necrosis consisted of individualized or clustered necrotic hepatocytes surrounded by few mononuclear inflammatory cells and rare neutrophils in all horses, apart from one SCID foal. Interestingly, [24] also tested SCID foals to determine whether the immune system could be involved in tissue damage: those foals resulted in lower viremia levels compared to immunocompetent animals and also liver enzymes could hardly be comparable with those of adult horses (even though the authors also question the reliability of detected GGT and SDH levels in foals, since it has been reported that within the first year of life those enzymes are susceptible to physiological variations), providing potential evidence of the role of T-cell-mediated immunopathology in EqHV infections.

At least 8 WPI, seroconversion could be observed and it overlapped with the elevation in liver enzymes, suggesting even more that the responses from the immune system have a function in determining liver damage. In addition, the rise of Ab levels usually seems to precede the drop in viral load, hinting that the immune system is also probably responsible for mechanisms of viral replication control; apart from those animals that evolved a chronic infection and could not clear it until 60 WPI, the average development of an acute EqHV infection could be described. Overall, the rate of acute infection in horses is more common than in humans, where the long-lasting chronic disease is the main outcome. However, it has not yet been reported if even acute EqHV infection in horses, especially racehorses and animals with high economical value, impact their wellbeing in terms of fatigue, poor performance, loss of appetite, or lethargy; hence, further studies are needed to pinpoint concurrent potential symptoms that could help to more quickly detect an EqHV infection.

### 7.2. Acute and Chronic Infection in Horses

Similar to human HCV infections, EqHV infections can be described as acute and chronic, depending on the duration of the viremia and the symptoms observed: usually, horses tend to develop acute infection more frequently compared to chronic disease and this is in contrast with what happens in humans. It was speculated that since a horse’s average life-span differs greatly from that of humans (horses live approximately 35 years), this could have led to a selective pressure on less-extensive liver injury during infection, making the acute outcome more common compared to the chronic one [19]. However, the reason behind the higher frequency of acute infection outcomes over chronic disease in horses is still unknown.

Originally, EqHV was not clearly associated with hepatotropism or hepatitis: in 2012, when EqHV had just been described [1], three EqHV positive horses in the UK were examined searching for signs of hepatitis or systemic disease, but no evidence for hepatic inflammation could be provided, apart from a slight increase in GGT levels in one horse; of note, another horse resulted viremic during the 5-month follow-up period, with liver enzyme levels more frequently to the upper end of the reference ranges, suggesting that EqHV could also establish persistent subclinical infections and low grade hepatitis in horses [33].

The very first report of natural potential connection between EqHV and hepatitis can be dated back to 2014, when a horse in Hungary showing severe jaundice was diagnosed with hepatitis and hepatic insufficiency, revealing to be positive for EqHV by conventional RT-PCR [18]. The animal had no history of transportation/transfusion/passive immunization, while the only treatments received were yearly vaccinations. EqHV was then ideally ascribed as the potential responsible for the hepatitis and liver damage, also because other pegiviruses possibly associated with liver infection (Theiler’s disease associated virus (TDAV) and Equine Pegivirus (EPgV)) were excluded.

Once parenteral infection was demonstrated to be a feasible route of transmission [11,17,19,21,24] fast improvements were made: kinetics of the infection were described and along with it, the general acute outcome of a EqHV infection: usually, it resolves in a few weeks, showing none to mild symptoms and it clears autonomously. Overall, the acute infection, if the subject is not screened for secondary purposes, might also go unnoticed, and being EqHV transmission routes are still elusive, this poses a risk of potential contagion, especially when horses are moved across stables or travel abroad for races/reproduction, making it easier to spread the disease worldwide.

Chronic disease, instead, has shown different and non-unanimous reports: there are several studies which describe the existence of a state of persistence of EqHV infection even over 6 months [11,19,20,26,38,72], sometimes reporting symptomatic behaviors of discomfort or illness [20,23,32,72], cases of co-infection of unknown effect [19,72], or presence of mycotoxins for which, however, related hepatotoxicity has not yet been reported [23]. Chronicity itself, then, still deserves a more in-depth study to be completely understood and to ascribe it exclusively to EqHV, beyond other possible or unknown causes. Being chronic the latter and rarer outcome for a EqHV infection, reliable and comparable data is often lacking, and therefore a wider sampling worldwide is auspicious along with a more accurate description of these few cases. This is of high relevance considering that the chronic development of the infection seems to pose a tangible risk for animals health: in fact, there have been reports of euthanization due to the too compromised condition of horses [19,23,72], especially related to liver failure, and this should not be overlooked, even if the infection usually has a subclinical development and is mainly silent.

### 7.3. Infection Steps: Protection against Reinfection

Humans are susceptible to HCV reinfection even after a curative therapy, especially people highly exposed to risk, suggesting that natural immunity against the virus could be short-lived [73]. Considering the high similarities HCV and EqHV share, some authors concentrated on whether also horses could be exposed to reinfection.

Studies highlight the fact that an immune mechanism that contributes to protection exists and there is a high chance that, in the future, these data could also lead to the development of a human HCV vaccine; however, further studies are hampered by the impossibility to set up a culture system for EqHV at the moment, making it harder for effective improvements to be made [17,21].

## 8. Cross-Species Transmission

Novel hepaciviruses have been reported in several animals, such as gerbils, ducks, sloths, and even marine species, such as the graceful catshark, *Proscyllium habereri,* demonstrating a large range of distribution worldwide, potentially originated from the flavi-like viruses found in invertebrates, which could represent the ancestral forms of modern vertebrate-infecting viruses [74,75,76,77].

Rodents and bats have been highlighted as a natural major reservoir of hepaciviruses due to their great distribution worldwide, their population numbers and the genetic diversity they show [14,40,43,78]. A recent study seems to corroborate the possibility that hepaciviruses can also spill over from rodents: in fact, the newly discovered Hepacivirus in sloths has its origin most strikingly from rodents [75]. If the discovery is solid, it could provide important evidences for the relevance of rodents in the spread of viruses in other species, including the potential origins of human HCV [14,40,43,75].

Another interesting cross species event that has been recently discovered refers to the Bovine Hepacivirus (Hepacivirus N) identified in wild boars, screened through a nested pan-Hepacivirus PCR. This highlights even more clearly that the range of hepaciviruses might be more extended than previously demonstrated [79].

### EqHV Cross-Species Transmission

Back to Equine Hepacivirus, to date only two studies have tried to provide data for the possible cross-species infection of EqHV from horses to humans [26,70]. The close relationship between HCV and Equine hepaciviruses could, in fact, reflect the ecological link between humans and horses, due to their primary use up until the twentieth century. This might have increased the chance of direct cross-species transmission, or through intermediate hosts: however, no evidence of zoonosis has been recorded and it seems, then, that despite any similitude, EqHV has its favorable host in horses [80].

Nonetheless, a few studies reported the presence of EqHV positive dogs in shared environments with positive horses, suggesting that cross-species events concerning EqHV may have occurred [38,39]. Lastly, it has been also reported how EqHV seems to be equally adapted to horses and donkeys [81], with the only difference that the latter seem to show enhanced virus clearance compared to horses.

## 9. Laboratory Diagnosis and Therapy

### 9.1. Laboratory Diagnosis

Actually, EqHV is not present in routinary laboratory diagnosis. The protocols present in literature were employed for prevalence studies or genetic characterization, and none of them has been validated for diagnostic purposes, according to international guidelines as those of WOAH [82,83].

#### 9.1.1. Serological Methods

At the moment, no commercial ELISA kits are available to screen anti EqHV antibodies or EqHV antigens in horse sera. Literature reports three main serological assays: the Luciferase immunoprecipitation system (LIPS), the Gaussia luciferase immunoprecipitation system (GLIPS), and western blotting. LIPS for EqHV [1,19,38,42,46,49] is a method that measures the luminescence emitted by Renilla luciferase fused to the EqHV protease/helicase NS3 antigen and allows the quantification of virus-specific serum antibodies; GLIPS [39,54] follows the same method but uses the Gaussia luciferase instead of the Renilla luciferase: the target antigen is still NS3. Lastly, western blotting [53,57] is used to detect antibodies against an EqHV core protein recombinant antigen bounded to a membrane.

#### 9.1.2. Biomolecular Methods

Biomolecular investigations on EqHV are conducted mainly though Real-Time PCR [1,33,44,46,47,49,54,61] and nested-PCR [1,32,33,49,50,52,54,55,56,60], targeting conserved portions of the genome as the 5′ UTR, NS3, and NS5B for the detection of viral RNA.

#### 9.1.3. Other Methods

As already reported, successful culture systems for EqHV have not yet been developed [17,21].

### 9.2. Therapy

Similarly as for validated diagnostic assays, specific therapeutic protocols are not available for EqHV, sinceknowledge of this infection is not yet widespread in the practitioners’ community.

## 10. Phylogenetic Analysis

Phylogenetic analysis on EqHV sequences concentrated on different genetic targets in order to provide a comprehensive and definitive description of the evolutionary history of the virus: to date, EqHV results to be highly conserved and it isolates in 1 unique genotype and 3 different subtypes (Subtype 1–3) distributed worldwide with no apparent preferential patterns [58,59].

The main target sequences that have been confronted are the 5′ UTR, NS3, NS5B, and E1/E2.

Already in 2015, a few studies highlighted the presence of two potential different groups of EqHV worldwide, confronting NS3 and NS5B sequences [51,54,84]. In 2017, these two major lineages were further explored and eventually characterized as two subtypes, using both concatenated sequences (5′ UTR + NS3 + NS5B) and partial genomic sequences (5′ UTR, NS3, NS5B), obtained from French strains and GenBank sequences [45].

In 2019, Lu et al. [58] proposed a new classification of EqHV that pointed out the existence of 3 subtypes, through the confrontation of the nearly complete polyprotein gene sequences. Secondarily, they eventually found evidence that in subtype 1 (American strains) recombination events had occurred within the NS5A and NS5B genes: even if recombination events in RNA viruses have been reported to sometimes influence viral replication and pathogenicity, the possibility that this event influences replication and pathogenicity in EqHV needs to be further researched.

Herein, a comprehensive overview of the EqHV (Hepacivirus A) genetic difference in horses and in all 14 known species of the *Hepacivirus* genus is presented.

Five targets have been chosen to provide a thorough state of the art of the *Hepacivirus* genome’s distribution worldwide and sequences were downloaded using the dataset from GenBank. The research for the EqHV sequences comprised the key words “((horse OR equine) AND Hepacivirus)” along with the target of interest “((5′ UTR) or (NS3) or (NS5B) or (Full genome))”; whether in case of the A-N Hepacivirus sequences the research focused on the species as proposed by [3], therefore: “((A) or (B) or (C) or (D) or (E) or (F) or (G) or (H) or (I) or (J) or (K) or (L) or (M) or (N) AND Hepacivirus))”. In both cases, the chosen sequences were those considered as the most informative because of their geographic representativeness and repeatability throughout several other publications. In the end, the sequences pinpointed as of interest were the Equine Hepacivirus 5′ UTR, NS3, and NS5B fragment, together with the EqHV Full Genome sequences and the NS5B fragment of the *Hepacivirus* A-N specimens.

The downloaded sequences, once isolated by target, were analyzed through BioEdit Sequence Alignement Editor (version 7.2.5, Tom Hall, 1997, Raleigh, NC, USA) and aligned through the ClustalW Multiple Alignment internal tool.

The sequences that did not align or that resulted too short to be confronted with the majority of the others were discarded, together with similar sequences that belonged to multiple submissions within the same publication: this was in order to guarantee balance among sequences and their geographical distribution. The target sequences were approximately 210 bp, 610 bp, 1710 bp, 8200 bp, and 2100 bp for respectively EqHV 5′ UTR, EqHV NS3, EqHV NS5B, EqHV Full Genome, and the NS5B fragment of the *Hepacivirus* A-N specimens.

The obtained sequences were then used to build the trees using the Maximum Likelihood method and Tamura–Nei model: the evolutionary analyses were conducted in MEGA11: Molecular Evolutionary Genetics Analysis (version 11.0.13, Tamura, Stecher, and Kumar 1993,Tokyo, Japan) with a bootstrap value of 1000. Alignments were verified for their statistical significance and robustness by Likelihood-mapping Analysis (LMA) using Tree-Puzzle software (version 5.2, Schmidt and von Haeseler, 2005, Vienna, Austria) [85] and each phylogenetic tree topology was verified using the Bootstrap method producing 1000 replicates [86]. Moreover, to confirm the clusters obtained, the CD-HIT-EST program (http://weizhongli-lab.org/cd-hit/ accessed on 1 August 2022) was employed [87]. For a further verification of the reliability of the results, the sequence position of the sequence from Burbelo et al., 2012 [1] were compared in their respective trees.

For the EqHV 5′ UTR tree, sequences already included in the trees displayed in Elia et al., 2017 [44] and Lu et al., 2016 [55] were used. Sequences deposited on Genbank as KX056116, KX056117 [56], KT175006, KT175011, KT175014, KT175033, KT175034, KT175036, KT175037, KT175040, KX239329, KX239337, KX239338, KX239349 [45], MH027992, MH027996, MH028000, MH028001, MH028004, MH028005 [47], LC440470, LC440466 [26], and MZ274312 (GenBank direct submission 2021) were included because they were not present in any previously published tree related to the 5′ UTR fragment. Sequence NC021153 [88], belonging to the Rodent Hepacivirus, was included as an outgroup.

In Figure 6, the phylogenetic tree is presented, together with the result of the analysis with Tree-Puzzle software (version 5.2, Schmidt and von Haeseler, 2005, Vienna, Austria).

For the EqHV NS3 tree, sequences already included in the trees displayed in Badenhorst 2018 [49], Paim 2019 [35], Schlottau 2019 [47], and Date 2020 [26] were included. Sequences deposited on Genbank, as MK644936 [58], MN734124 [59], MT955622 (GenBank direct submission 2021), and MZ274312 (GenBank direct submission 2021) were included because they were not present in any previously published tree related to the NS3 fragment. Sequence KC411778 [43], belonging to the Rodent Hepacivirus, was included as an outgroup. In Figure 7, the phylogenetic tree is presented, together with the result of the analysis with Tree-Puzzle software (version 5.2, Schmidt and von Haeseler, 2005, Vienna, Austria).

For the EqHV NS5B tree, sequences already included in the trees displayed in Figuereido 2018 [52], Pronost 2019 [28], and Date 2020 [26] were included. Sequences recently deposited on Genbank, as MH027992, MH027996, MH027999, MH028004, MH028007 [47], MK284504 (2018), MK644937 [58], MN734124 [59], MT955622 (GenBank direct submission 2021), and MZ274312 (GenBank direct submission 2021) were included because they were not present in any previously published tree related to the NS5B fragment. Sequence NC021153 [88], belonging to the Rodent Hepacivirus was included as an outgroup. In Figure 8 the phylogenetic tree is presented, together with the result of the analysis with Tree-Puzzle software (version 5.2, Schmidt and von Haeseler, 2005, Vienna, Austria).

For the EqHV Full Genome tree, sequences already included in the trees displayed in Tegtmeyer 2019 [23], Lu 2019 [58], Date 2020 [26], and Wu 2020 [59] were included. Sequences recently deposited on Genbank as MZ274312 (GenBank direct submission 2021) and MT955622 (GenBank direct submission 2021) were included because they were not present in any previously published tree. Sequence NC021153 [88], belonging to the Rodent Hepacivirus, was included as an outgroup. In Figure 9, the phylogenetic tree is presented, together with the result of the analysis with Tree-Puzzle software (version 5.2, Schmidt and von Haeseler, 2005, Vienna, Austria).

For the NS5B Hepacivirus A-N tree, sequences already included in the trees displayed in Paim et al., 2019 [35] and Schlottau et al., 2019 [47] to create the NS3 tree and An et al., 2022 [76] were included. Sequences recently deposited on Genbank, as MK737639 (Duck Hepacivirus) [74]; KR902729 (Wenling Shark) [77]; MH844500 (Sloth Hepacivirus), and MG211815.1 (called in a direct submission on GenBank as Hepacivirus P) [75]; MT955622 (GenBank direct submission 2021); MK284504 (GenBank direct submission 2018) were included as they were not present in any previously published NS5B Hepacivirus A-N tree. Two Human Hepatitis C virus sequences (KY620564 and AB049088) were included because they were not present in any previously published tree [89,90]. The sequences MW853928 (GenBank direct submission 2021), belonging to Classical Swine Fever, and MN792937 [91], belonging to West Nile Virus, were included as outgroups. In Figure 10 the phylogenetic tree is presented, together with the result of the analysis with Tree-Puzzle software (version 5.2, Schmidt and von Haeseler, 2005, Vienna, Austria).

## 11. EqHV as a Model for HCV

The World Health Organization (WHO) estimated that in 2019 approximately 58 million people were globally chronically infected with HCV, with approximately 1.5 million new infections occurring per year and over 290,000 deaths in 2019, predominantly associated with cirrhosis and hepatocellular carcinoma (HCC) due to HCV infection [92].

Historically, HCV research relied on chimpanzees, however, after 2011, globally increased restrictions on the use of chimpanzees in biomedical research for ethical reasons greatly hampered any improvements and genetically humanized mouse models became central for discovery efforts due to their more direct application to human infection [93].

The discovery of a homolog virus of HCV became a turning point in human hepaciviruses studies: since its first report in 2011 and its genetic and biological characterization in the following years, EqHV nurtured expectations, possibly providing a better understanding of the human disease through a new in vivo model, with implication for research into the pathogenesis, prevention, and treatment of infection, eventually and possibly leading to a vaccine therapy [1,8,17].

Ramsay et al., 2015 [24] officially proposed the horse as a surrogate model for HCV research, underlining several advantages: long life span to study chronicity over several years, a large hepatic mass necessary to do serial liver biopsies and, lastly, the opportunity to study the closest known relative of HCV in its natural host. As a matter of fact, an essential step toward an animal model for hepatitis is the identification of a tractable, long-living host with biological mechanisms similar to human HCV infections and pathogenesis of the virus, and since horses have provided data for chronic infections from EqHV, they could make a good candidate [11,14,19,20,23,26,38,52,72]. Moreover, Ramsay et al. [24] stated that the results they obtained in their study are comparable with those obtained with HCV in humans and chimpanzees, making the horse a convincing translational animal model for HCV.

However, despite these promising overlaps, the chance that horses could lead to a significant twist in HCV discoveries has been recently questioned, since natural clearance in horses highlights completely different immune responses compared to humans, complicating the assessment of mechanisms responsible for chronicity and, therefore, the assessment of a potential vaccine [94].

## 12. Discussion and Conclusions

The coronavirus pandemic showed us how zoonotic pathogens can have a disastrous and threatening impact on human health worldwide: its fast emergence underlined the importance of monitoring both livestock and wildlife as much as possible, while searching for new animal viruses, but also feasible animal models that could help us deepen our knowledge on renowned viruses and widespread pathologies, such as human hepatitis. Both homologies and differences can be informative, in fact, in order to better understand unknown diseases or pathologies that were believed to affect only humans.

EqHV is one among the newly discovered viruses in equine medicine, and it has been identified as the closest known relative of human HCV [1]. For these two reason an extensive review is presented in this paper as a benchmark of the knowledge of this virus, including a meta-analysis of the serological and biomolecular prevalence worldwide. Moreover, an updated phylogenetical analysis is presented for the three main segments investigated in literature (5′ UTR, NS3, NS5B) and for the EqHV full genome, together with an update of the sequences of the whole *Hepacivirus* genus, with recently deposited sequences.

The review and the resuming of all the epidemiological features together with risk factors could be a valid guide to prevent infection and to detect it in a herd, being actually surely underestimated in its prevalence.

The meta-analysis for serological prevalence resulted in an estimated medium prevalence around the world of 47.11% [CI 95% 32.38–61.83]. The analysis detected heterogeneity among the studies, even though the statistical test for funnel plot asymmetry resulted not significant. The selection model and the RoBMA confirmed the presence of heterogeneity, while publication bias resulted not significant (*p* = 0.501 for selection model, *p* = 0.561 for PET model and BF = 1.606, thus <10 for RoBMA). Analysis for area, country and species revealed a significant heterogeneity among the three areas investigated (Wald test *p*-value Asia = 3.3 × 10^−4^, North America = 0.014, and Europe = 0.009).

The meta-analysis for biomolecular prevalence resulted in an estimated medium prevalence around the world of 6.95% [CI 95% 5.41–8.50]. The analysis detected heterogeneity among the studies, even though the statistical test for funnel plot asymmetry resulted discordant with the rank correlation test not significant, even with a *p* = 0.069 and the Egger’s test strongly significant. The selection model and the RoBMA confirmed the presence of heterogeneity, while publication bias resulted not significant (*p* = 0.427 for selection model and BF = 0.289, thus <10 for RoBMA). Analysis for area, country, and species revealed a significant heterogeneity among the three areas investigated, in particular the prevalence in Europe (*p* = 0.012 for the Wald test) seems to be lower than for the other continents. This datum is in concordance with the results of [95].

The results of phylogenetic analysis on the three main segment and on the full genome of eqHV resulted robust for several reasons: the sum of vertexes of the Tree puzzle analysis was always higher than 80%; the sequence included as controls always clustered in the same subtype as in the original publication [58], the bootstrap values are mostly satisfactory, and the result of CD-HIT-EST confirmed the clusters emerged in the phylogenetic trees. Having said that, the virus seems to have a world wide spread with no clustering of the subtype in any particular region; the new sequences deposited and included indeed clustered in all three subtypes, further confirming results of previous studies [58,59].

The NS5B tree for the whole genus *Hepacivirus* confirms the similarity among the EqHV and the HCV. Furthermore, several newly discovered hepaciviruses have been included in the phylogenetic analysis as the wenling shark virus, that clusters with Hepacivirus J; the duck hepacivirus, that clusters with Hepacivirus D; the sloth hepacivirus, clustering together with Hepacivirus P, as presented by [75], and resulted similar to Hepacivirus E, F, G, and H; and, finally, the gerbil hepacivirus that clusters with Hepacivirus E and that affects rodent as already presented by [76].

In conclusion this review provides a valuable reference guide for practitioners and for researchers in conducting further study to confirm the data or to deeply explore some of the characteristic investigated but not completely assessed.

## Figures and Tables

**Figure 1 animals-12-02486-f001:**
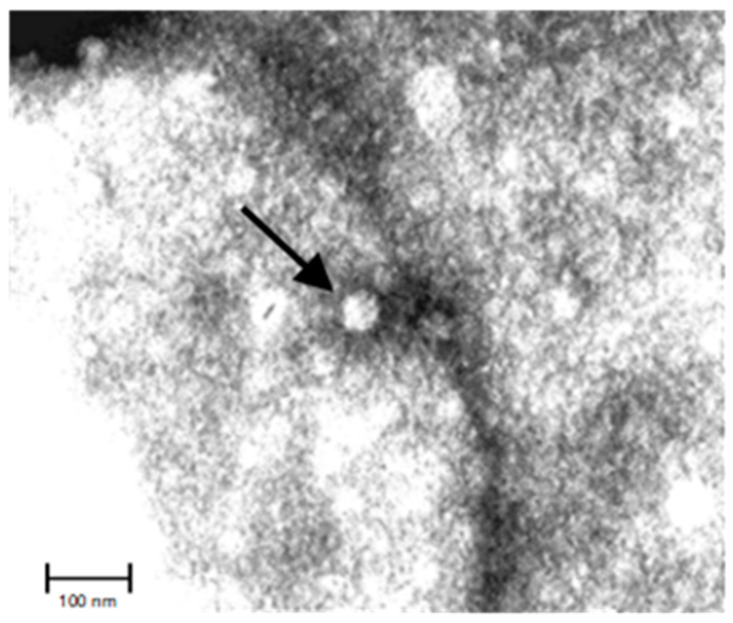
Negative staining electron microscopy picture of a virion detected in a EqHV Real Time PCR positive serum sample belonging to the National Reference Center for Equine Diseases.

**Figure 2 animals-12-02486-f002:**
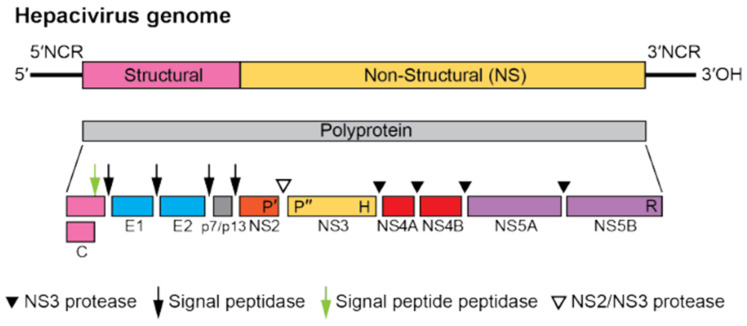
Hepacivirus genome organization (not to scale) and polyprotein processing. Picture from [12].

**Figure 3 animals-12-02486-f003:**
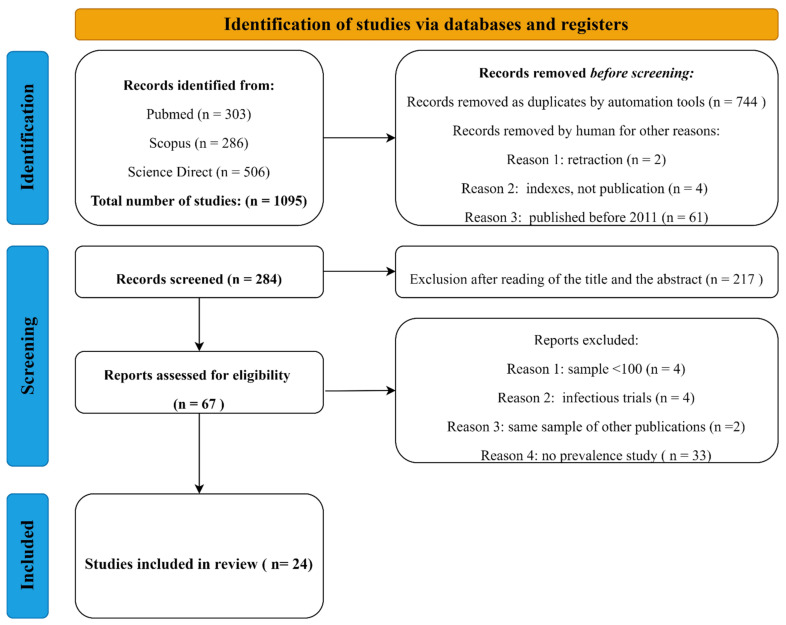
PRISMA flowchart displaying the selection process for the articles employed for the meta-analysis.

**Figure 4 animals-12-02486-f004:**
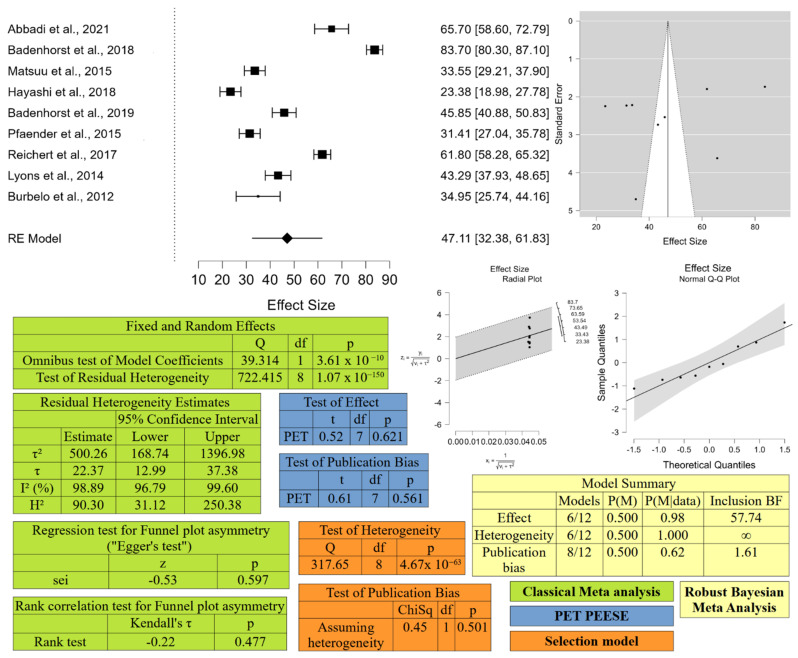
Meta-analyses result for serological prevalence of EqHV conducted with JASP (version 0.16.3, University of Amsterdam, Amsterdam, The Netherlands). At the upper left corner, the forest plot is displayed, at the upper right corner the diagnostic plots (Funnel plot, Radial plot, and Normal Q-Q plot). In the bottom half of the figure, results of Classical Meta-analysis, PET PEESE analysis, Selection Model, and Robust Bayesian Meta-analysis are presented. BF: Bayes factors; ChiSq: Chi Square value; df: degrees of freedom; H2: value of H2 statistic for assessing heterogeneity; I2: value of I2 statistic for assessing heterogeneity; Kendall’s τ: value of Kendall rank correlation coefficient; p: *p*-value; P(M) prior probability; P(M|data): posterior probability; Q: Cochran’s Q statistic; sei: vector with the corresponding standard errors; t: t-statistics value; τ: standard deviation of the true effect sizes; τ2: variance of the true effect sizes; z: z-statistic value. The mentioned references are [1,19,38,39,42,46,49,54,57].

**Figure 5 animals-12-02486-f005:**
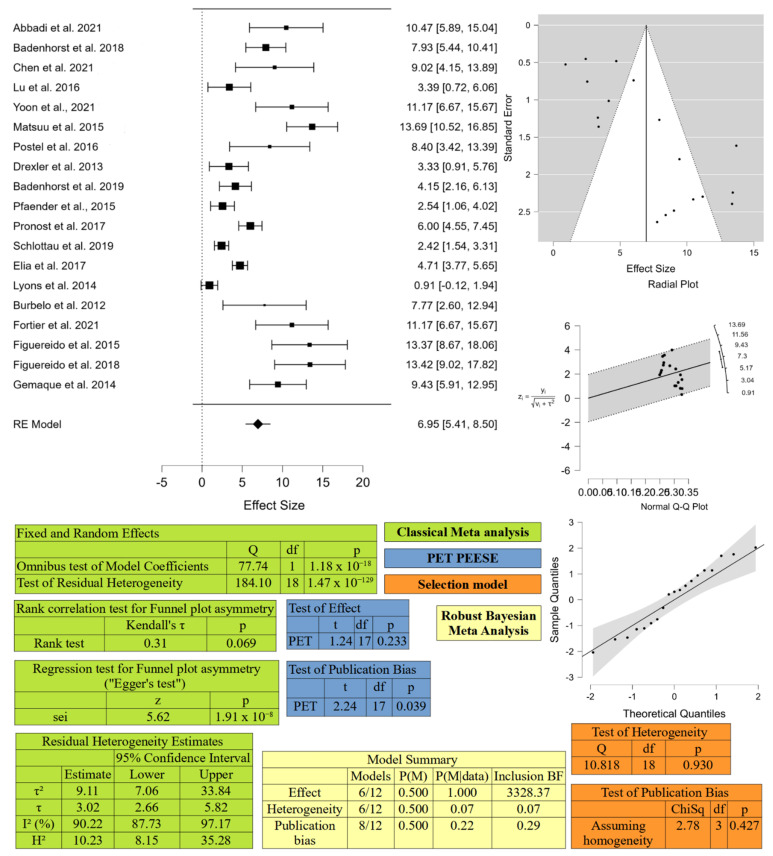
Meta-analysis results for virological prevalence of EqHV conducted with JASP (version 0.16.3, University of Amsterdam, Amsterdam, The Netherlands). At the upper left corner, the forest plot is displayed, at the upper right corner the diagnostic plots (Funnel plot, Radial plot, and Normal Q-Q plot). In the bottom half of the figure, results of Classical Meta-analysis, PET PEESE analysis, Selection Model, and Robust Bayesian Meta-analysis are presented. BF: Bayes factors; ChiSq: Chi Square value; df: degrees of freedom; H2: value of H2 statistic for assessing heterogeneity; I2: value of I2 statistic for assessing heterogeneity; Kendall’s τ: value of Kendall rank correlation coefficient; p: *p*-value; P(M) prior probability; P(M|data): posterior probability; Q: Cochran’s Q statistic; sei: vector with the corresponding standard errors; t: t-statistics value; τ: standard deviation of the true effect sizes; τ2: variance of the true effect sizes; z: z-statistic value. The mentioned references are [1,19,32,34,38,39,42,43,44,45,47,49,50,51,52,54,55,60,61].

**Figure 6 animals-12-02486-f006:**
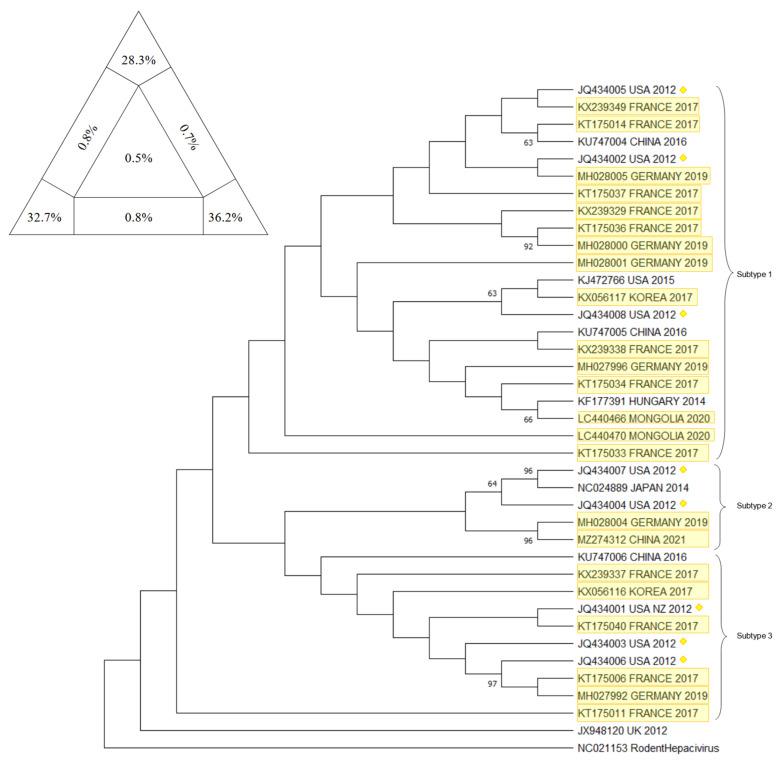
The evolutionary tree for 5′ UTR fragment of EqHV inferred by using the Maximum Likelihood method and the Tamura–Nei model. The tree with the highest log likelihood (−866.98) is shown. The percentage of trees in which the associated taxa clustered together is shown next to the branches. Initial tree(s) for the heuristic search were obtained automatically by applying Neighbor-Join and BioNJ algorithms to a matrix of pairwise distances estimated using the Tamura–Nei model, and then selecting the topology with a superior log likelihood value. This analysis involved 39 nucleotide sequences. There were 209 positions in the final dataset. Evolutionary analyses were conducted in MEGA11: Molecular Evolutionary Genetics Analysis (version 11.0.13, Tamura, Stecher, and Kumar 1993,Tokyo, Japan). Yellow rectangle frames are the sequences included that have never been included before in a tree. Yellow diamonds identify the sequence from Burbelo et al. (2012) [1], included to verify the presence of 3 subtypes. At the upper right corner, results of Tree-Puzzle software (version 5.2, Schmidt and von Haeseler, 2005, Vienna, Austria) analysis are displayed. Sum of vertexes is 97.2%.

**Figure 7 animals-12-02486-f007:**
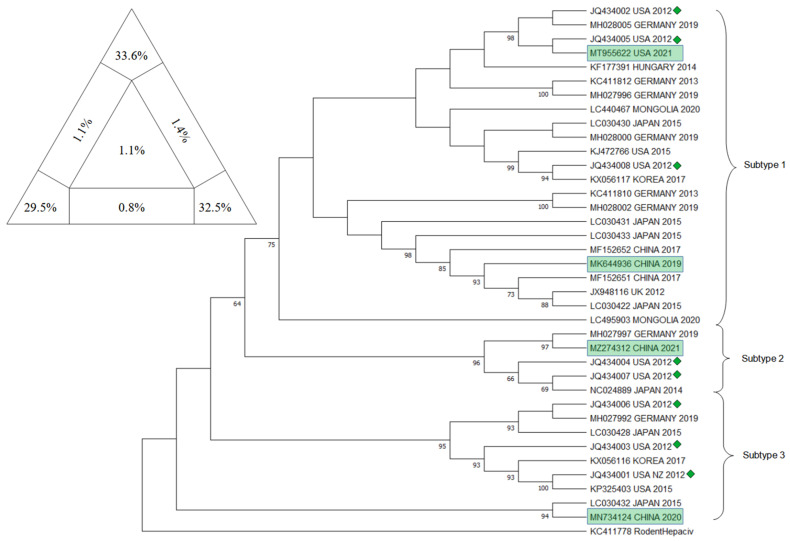
The evolutionary tree for NS3 fragment of EqHV inferred by using the Maximum Likelihood method and Tamura–Nei model. The tree with the highest log likelihood (−4856.54) is shown. The percentage of trees in which the associated taxa clustered together is shown next to the branches. Initial tree(s) for the heuristic search were obtained automatically by applying Neighbor-Join and BioNJ algorithms to a matrix of pairwise distances estimated using the Tamura–Nei model, and then selecting the topology with superior log likelihood value. This analysis involved 38 nucleotide sequences. There were 604 positions in the final dataset. Evolutionary analyses were conducted in MEGA11: Molecular Evolutionary Genetics Analysis (version 11.0.13, Tamura, Stecher, and Kumar 1993,Tokyo, Japan). Green rectangle frames are the sequences included that have never been included before in a tree. Green diamonds identify the sequence from Burbelo et al. (2012) [1], included to verify the presence of 3 subtypes. At the upper right corner, results of Tree-Puzzle software (version 5.2, Schmidt and von Haeseler, 2005, Vienna, Austria) analysis are displayed. Sum of vertexes is 95.6%.

**Figure 8 animals-12-02486-f008:**
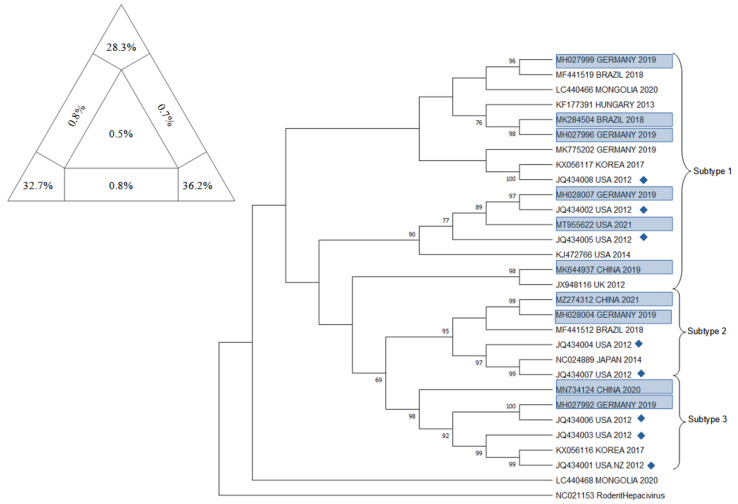
The evolutionary tree for NS5B fragment of EqHV inferred by using the Maximum Likelihood method and Tamura–Nei model. The tree with the highest log likelihood (−14,989.04) is shown. The percentage of trees in which the associated taxa clustered together is shown next to the branches. Initial tree(s) for the heuristic search were obtained automatically by applying Neighbor-Join and BioNJ algorithms to a matrix of pairwise distances estimated using the Tamura–Nei model, and then selecting the topology with superior log likelihood value. This analysis involved 30 nucleotide sequences. There were 1680 positions in the final dataset. Evolutionary analyses were conducted in MEGA11: Molecular Evolutionary Genetics Analysis (version 11.0.13, Tamura, Stecher, and Kumar 1993,Tokyo, Japan). Blue rectangle frames are the sequences included that have never been included before in a tree. Blue diamonds identify the sequence from Burbelo et al. (2012) [1], included to verify the presence of 3 subtypes. At the upper right corner, results of Tree-Puzzle software (version 5.2, Schmidt and von Haeseler, 2005, Vienna, Austria) analysis are displayed. Sum of vertexes is 97.2%.

**Figure 9 animals-12-02486-f009:**
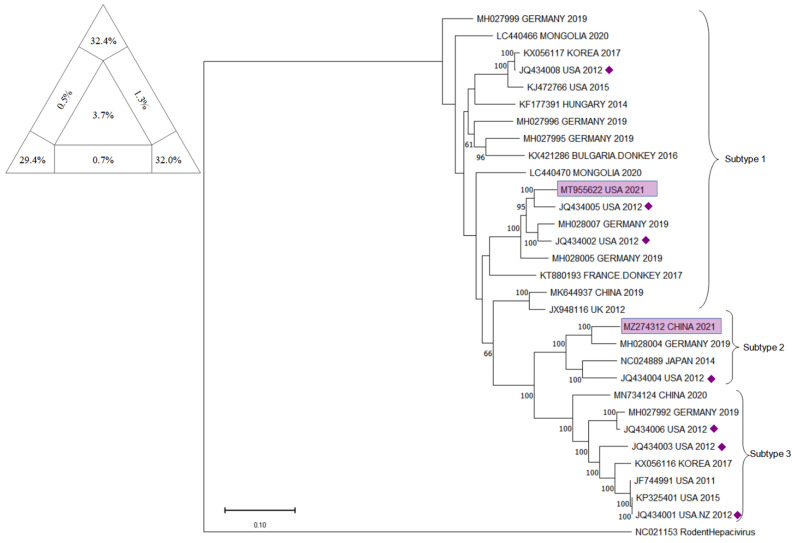
The evolutionary tree for EqHV full genome inferred by using the Maximum Likelihood method and Tamura–Nei model. The tree with the highest log likelihood (−75,419.31) is shown. The percentage of trees in which the associated taxa clustered together is shown next to the branches. Initial tree(s) for the heuristic search were obtained automatically by applying Neighbor-Join and BioNJ algorithms to a matrix of pairwise distances estimated using the Tamura–Nei model, and then selecting the topology with superior log likelihood value. The tree is drawn to scale, with branch lengths measured in the number of substitutions per site. This analysis involved 31 nucleotide sequences. There were 7700 positions in the final dataset. Evolutionary analyses were conducted in MEGA11: Molecular Evolutionary Genetics Analysis (version 11.0.13, Tamura, Stecher, and Kumar 1993,Tokyo, Japan). Purple rectangle frames are the sequences included that have never been included before in a tree. Purple diamonds identify the sequence from Burbelo et al. (2012) [1], included to verify the presence of 3 subtypes. At the upper right corner, results of Tree-Puzzle software (version 5.2, Schmidt and von Haeseler, 2005, Vienna, Austria) analysis are displayed. Sum of vertexes is 93.8%.

**Figure 10 animals-12-02486-f010:**
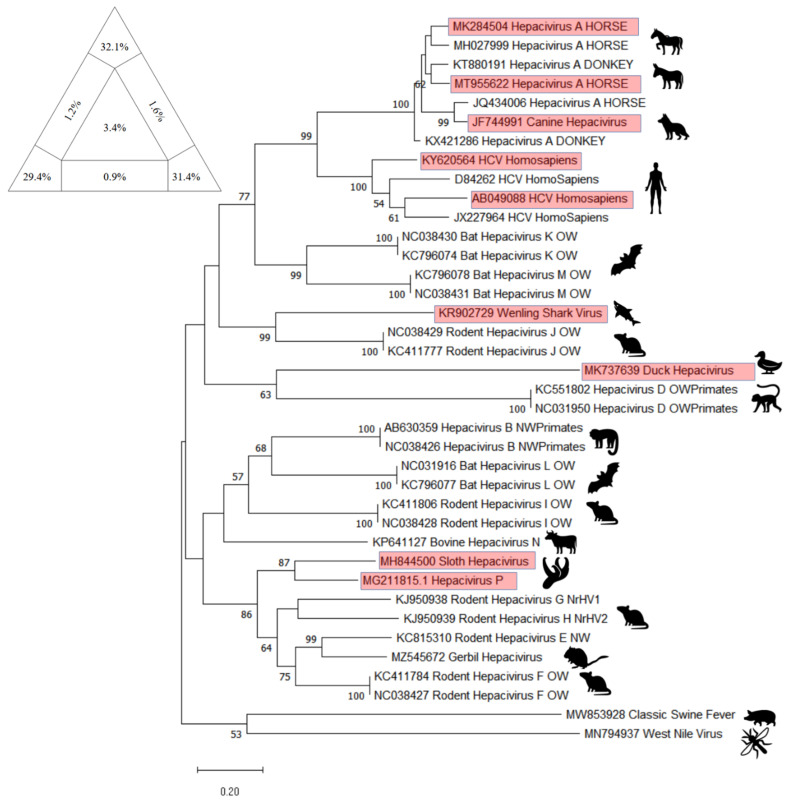
The evolutionary tree of NS5B fragment for Hepacivirus genus inferred by using the Maximum Likelihood method and Tamura–Nei model. The tree with the highest log likelihood (−38,096.60) is shown. The percentage of trees in which the associated taxa clustered together is shown below the branches. Initial tree(s) for the heuristic search were obtained automatically by applying Neighbor-Join and BioNJ algorithms to a matrix of pairwise distances estimated using the Tamura–Nei model, and then selecting the topology with superior log likelihood value. The tree is drawn to scale, with branch lengths measured in the number of substitutions per site. This analysis involved 38 nucleotide sequences. There were 1373 positions in the final dataset. Evolutionary analyses were conducted in MEGA11: Molecular Evolutionary Genetics Analysis (version 11.0.13, Tamura, Stecher, and Kumar 1993,Tokyo, Japan). Red rectangle frames are the sequences included that have never been included before in a tree. At the upper right corner, results of Tree-Puzzle software (version 5.2, Schmidt and von Haeseler, 2005, Vienna, Austria) analysis are displayed. Sum of vertexes is 92.9%.

**Table 1 animals-12-02486-t001:** Comparison of actual and previous classifications of *Hepacivirus* genus. Modified by: Smith, Donald B., et al. [3] OW: Old World; NW: New World; HV: Hepacivirus; NPHV: Nonprimate Hepacivirus; EqHV: Equine Hepacivirus.

**Hepacivirus Species**	**A**	**B**	**C**	**D**	**E**	**F**	**G**
Previous identifier	Canine HV, NPHV, EqHV	GBV-B	HCV	Guereza Hepacivirus	Rodent Hepacivirus	Rodent Hepacivirus	Norway Rat HV 1
Host	Horse (Dog?)	NW primates	Human	OW Primate	NW Rodent	OW Rodent	Global Rodent
**Hepacivirus species**	**H**	**I**	**J**	**K**	**L**	**M**	**N**
Previous identifier	Norway Rat HV 2	Rodent Hepacivirus	Rodent Hepacivirus	Bat Hepacivirus	Bat Hepacivirus	Bat Hepacivirus	Bovine Hepacivirus
Host	Global Rodent	OW Rodent	OW Rodent	OW bat	OW bat	OW bat	Cow

**Table 2 animals-12-02486-t002:** EqHV virological and serological prevalence data available in literature, divided for continent, country, and species. n.d.: not determined.

Continent	Country	Tested Species	Prevalences (Only Horses)		References
			PCR (%)	Serum (%)	
Europe	Italy	Horse, Donkey	91/1932 (4.7%)	n.d.	Elia et al., 2017 [44]
	Hungary	Horse	1/1 (100%)	n.d.	Reuter et al., 2014 [18]
	UK	Horse	3/142 (2.1%)	n.d.	Lyons et al., 2012 [33]
	UK	Horse, Donkey	3/328 (<1%)	142/328 (43.3%)	Lyons et al., 2014 [38]
	Germany	Horse	7/210 (3.3%)	n.d.	Drexler et al., 2013 [43]
	Germany	Horse	11/433 (2.5%)	136/433 (31.4%)	Pfaender et al., 2015 [19]
	Germany	Horse	10/119 (8.4%)	n.d.	Postel et al., 2016 [34]
	Germany	Horse	134/733 (18.2%)	453/733 (61.8%)	Reichert et al., 2017 [46]
	Germany	Horse	28/1155 (2.4%)	n.d.	Schlottau et al., 2019 [47]
	Austria	Horse	16/386 (4.15%)	177/386 (45.9%)	Badenorst et al., 2019 [42]
	Austria	Horse, Donkey	1/259 (0.38%)	n.d.	Badenorst et al., 2021 [48]
	France	Horse	69/1229 (5.6%)	n.d.	Pronost et al., 2016 [29]
	France	Horse, Thoroughbreds	62/1033 (6.2%)	n.d.	Pronost et al., 2017 [45]
Africa	South Africa	Horse, Thoroughbreds	36/454 (7.9%)	380/454 (83.70%)	Badenhorst et al., 2018 [49]
	Marocco	Horse	18/172 (10.50%)	113/172 (65.70%)	Abbadi et al., 2021 [39]
North America	US	Horse	8/103 (7.7%)	36/103 (34.9%)	Burbelo et al., 2012 [1]
	US	Horse	2/14 (14.28%)	n.d.	Tomlinson et al., 2019 [22]
South America	Brazil	Horse, Donkey, Mule	25/265 (9.4%)	n.d.	Gemaque et al., 2014 [50]
	Brazil	Horse, Donkey	27/202 (13.4%)	n.d.	Figuereido et al., 2015 [51]
	Brazil	Horse	31/231 (13.4%)	n.d.	Figuereido et al., 2018 [52]
Asia	China	Horse, Donkey, Mule	6/177 (3.4%)	n.d.	Lu et al., 2016 [55]
	China	Horse	6/13 (46.2%)	n.d.	Lu et al., 2019 [58]
	China	Horse, Warmblood	19/60 (3.2%)	n.d.	Wu et al., 2020 [59]
	China	Horse	12/133 (9%)	n.d.	Chen et al., 2021 [60]
	Mongolia	Horse	141/299 (47.1%)	n.d.	Date et al., 2020 [26]
	Japan	Horse	11/31 (35.6%)	7/31 (22.6%)	Tanaka et al., 2014 [53]
	Japan	Horse, Thoroughbreds	62/453 (13.7%)	152/453 (33.5%)	Matsuu et al., 2015 [54]
	Japan	Indigenous breed horses	n.d.	83/355 (23.9%)	Hayashi et al., 2018 [57]
	Korea	Horse, Thoroughbreds	14/74 (18.9%)	n.d.	Kim et al., 2017 [56]
	Korea	Horse	13/160 (8.1%)	n.d.	Yoon et al., 2021 [32]
Oceania	Australia	Horse	21/188 (11.2%)	n.d.	Fortier et al., 2021 [61]

**Table 3 animals-12-02486-t003:** Variation in biochemical liver functionality parameters reported in literature for EqHV positive subjects. Species investigated and reference intervals reported in each study is presented. n.d.: not determined.

References	Species	Altered Values	Reference Intervals
Lyons et al., 2012 [33]	Horse	↑ GGT	GGT < 40 U/mL
Ramsay et al., 2015 [24]	Horse	↑ GGT	GGT 14–40 U/L
		↑ SDH	SDH 4–14 U/L
Scheel et al., 2015 [11]	Horse	↑ GLDH	GLDH 1–8 U/L
		↑ GGT	GGT 8–29 U/L
		↑ AST	AST 199–374 U/L
		↑ SDH	SDH 0–11 U/L
Gather et al., 2016 [25]	Horse	↑ GLDH	GLDH < 6 U/L
		↑ GGT	GGT < 20 U/L
		↑ AST	AST < 170 U/L
Pfaender et al., 2017 [21]	Horse	↑ GLDH	GLDH < 6 U/L
		↑ GGT	GGT < 20 U/L
		↑ AST	AST < 170 U/L
Tomlinson et al., 2021 [17]	Horse	↑ GLDH	n.d.
		↑ GGT	n.d.
		↑ AST	n.d.
		↑ SDH	n.d.

## Data Availability

The datasets used and/or analyzed during the current study are available from the corresponding author on reasonable request.

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
