# Peer review of "Equine Hepacivirus: A Systematic Review and a Meta-Analysis of Serological and Biomolecular Prevalence and a Phylogenetic Update"

_animals, 2022, doi:10.3390/ani12192486_

Round 1

Reviewer 1 Report

The proposed paper is a very thorough overview of a virus, the equine hepacivirus, which has been relatively recently identified as the closest homologue to the human hepatitic C virus and benefited of an increased recognition of the health burden it induces in horses. The review, covering exclusive aspects of the infection, provides important information and also a foundation for EqHV disease diagnosis, prevention, and control, as tackled by research projects in development (Epidemiology, Transmission, and Pathogenicity of Equine Hepacivirus, Tomlinson et al., USDA (USDA-NIFA), 2022).

The review also places the EqHV in the taxonomic framework of the well-know, impacting group of zoonotic diseases’ agents, the Flaviviruses, thus raising the awareness towards new representatives of the group.

As the prevention of the disease leans on EqHV early identification, meta-analysis is used to present the geographical distribution, viral prevalence and sero-prevalence of the agent. 

Not only individual and herd risk factors are evaluated, but also the cross-species transmission and potential zoonotic impact are discussed. 

Accordingly, the paper meets the requirements (for the moment) for being a guide for practitioners but also researchers in the field and stressing the importance of preventive measures by highlighting the transmission routes.

I consider this review is a very thorough overview of the equine hepacivirus, approaching the EqHV from numerous angles: taxonomy, structure and replication, very comprehensive epidemiology (transmission routes, geographical distribution - EqHV virological and serological prevalence data available in literature), very clear and well-organised meta-analysis, risks, phylogenetic analysis and so on…

I would not consider this methodology needs to be improved.

By stating “The review and the resuming of all the epidemiological features together with risk 744 factors could be a valide guide to prevent the infection and detect it in a herd, being actu-745 ally surely understimated in its prevalence.” and “The NS5B tree for the whole genus Hepacivirus confirm the similarity among the 773 EqHV and the HCV.” I think the authors make their point towards pulling a strong warning signal on the importance of emerging zoonoses, and not only, since similarities with other animal viruses were identified (“Furthermore, several newly discovered hepaciviruses have been in-774 cluded in the phylogenetic analysis as the Wenling shark viruse, that clusters with Hepaci-775 virus J; the duck hepacivirus, that clusters with Hepacivirus D; the Sloth hepacivirus, clus-776 tering together with Hepacivirus P, as presented by [75], and resulted similar to Hepaci-777 virus E, F, G, and H; and finally the gerbil hepacivirus, that clusters with Hepacivirus E, 778 that affects rodent”).

The references strongly support the entire structure of the paper.

The phylogenetic analysis was supported by most of the figures (n=5), but the meta-analysis offers a very logical sequence of table-fig alternation, leading the reader to a better understanding of the distribution and importance of this emerging virus.

Author Response

Dear Reviewer 1,

thank you very much for your good evaluation.

Kind regards

Roberto Nardini

Reviewer 2 Report

The Authors carry out a review that presents current knowledge concerning EqHV infection in horses, including its epidemiology, evolutionary characteristics, diagnostics and prevention practices. A meta-analysis of serological and biomolecular prevalence and an updated phylogenetic description is also presented.

The title indicates the aim of the manuscript and the abstract clearly indicates the work objective.

The research design of the study is adequately described and results are clearly presented. The methodology is appropriate.

Manuscript is well-structured and scientifically sound therefore. The conclusions are consistent with the evidence and arguments presented.

I have a suggestion:

Please add “and” in the title after “Systematic Review” (Equine Hepacivirus: a Systematic Review and a Meta-Analysis of Serological and Biomolecular Prevalence and Phylogenetic Update)

I support the publication of this manuscript after the abovesaid minor revision.

Author Response

Dear Reviewer 2,

thank you very much for your evaluation and suggestion.

Please find in the revised manuscript the title modified according to your comment.

Kind regards,

Roberto Nardini

Reviewer 3 Report

A very good and comprehensive overview of the current state of knowledge on equine hepacivirus and will be welcomed by all those interested in viral infections in horses.

Minor notes:

Fig 1: which shows the EM negative staining of equine hepacivirus in serum, is of poor quality and adds little to the understanding of the structure of the virus - if no better EMimage is available, then it would be better to get permission to publish an EM-picture from another paper if possible - or live out showing an EM image

A paragraph break is missing above line 247

Chapter 9:  it would still be valuable for interested researchers to get more detailed information on current diagnostic methods, e.g. virus isolation, PCR and serological methods etc.

Author Response

Dear Reviewer 3,

thank you very much for your evaluation and suggestions to whose I respond below:

Fig 1: We are aware that the quality of the figure is not the best, although it is the best possible with the equipment available in our laboratory. We would strongly like to mantain it in the paper as it is the first real image of an EqHV virion. The picture available on other papers are reffered either to HCV or Flaviviridae in general.

A paragraph break is missing above line 247:

We are not sure if you referred to the line below the table, anyway we added a break to separate it from the table.

Chapter 9:  it would still be valuable for interested researchers to get more detailed information on current diagnostic methods, e.g. virus isolation, PCR and serological methods etc.

We revised the chapter as you can see in the revised paper from LINE 542-569.

We hope the revised manuscript would result eligible for publication.

Kind regards,

Roberto Nardini